# Perspectives for Improving the Tumor Targeting of Nanomedicine via the EPR Effect in Clinical Tumors

**DOI:** 10.3390/ijms241210082

**Published:** 2023-06-13

**Authors:** Jinseong Kim, Hanhee Cho, Dong-Kwon Lim, Min Kyung Joo, Kwangmeyung Kim

**Affiliations:** 1Graduate School of Pharmaceutical Sciences, College of Pharmacy, Ewha Woman’s University, Seoul 03760, Republic of Korea; wlstjd233038@gmail.com (J.K.); ricky@ewha.ac.kr (H.C.); 2KU-KIST Graduate School of Converging Science and Technology, Korea University, Seoul 02841, Republic of Korea; dklim@korea.ac.kr; 3Noxpharm Co., Ltd., #518, 150, Bugahyeon-ro, Seodaemun-gu, Seoul 03759, Republic of Korea

**Keywords:** nanomedicine, EPR effect, tumor microenvironments, clinical translation

## Abstract

Over the past few decades, the enhanced permeability and retention (EPR) effect of nanomedicine has been a crucial phenomenon in targeted cancer therapy. Specifically, understanding the EPR effect has been a significant aspect of delivering anticancer agents efficiently to targeted tumors. Although the therapeutic effect has been demonstrated in experimental models using mouse xenografts, the clinical translation of the EPR effect of nanomedicine faces several challenges due to dense extracellular matrix (ECM), high interstitial fluid pressure (IFP) levels, and other factors that arise from tumor heterogeneity and complexity. Therefore, understanding the mechanism of the EPR effect of nanomedicine in clinics is essential to overcome the hurdles of the clinical translation of nanomedicine. This paper introduces the basic mechanism of the EPR effect of nanomedicine, the recently discussed challenges of the EPR effect of nanomedicine, and various strategies of recent nanomedicine to overcome the limitations expected from the patients’ tumor microenvironments.

## 1. Introduction

Over the past several decades, anticancer agents have been developed to improve efficiency while minimizing side effects. Tumor targeting therapies, such as nanomedicine transportation, have also been considered as therapeutic methods that address the limitations of conventional chemodrugs, including lack of specificity and biodistribution. Moreover, nanomedicine-based therapies for tumor targeting have provided clinicians with the ability to meet the unmet needs of current cancer treatments. Therefore, understanding the pathophysiological characteristics of solid tumor growth is crucially important. To achieve effective transportation of anticancer agents to the tumor site via passive targeting, various methods have been applied. One promising method is the EPR effect, which specifically arises from tumor tissue. It is already known that tumor vessels are highly permeable to macromolecular compounds (typically nanoparticles, liposomes, and macromolecular drugs), and after these macromolecular compounds enter the tumor tissue, they will be trapped inside the tumor tissue for a prolonged period. Additionally, dysfunctional lymphatic drainage compared to normal tissues helps the selective accumulation of the macromolecular compounds. This phenomenon is referred to as the EPR effect, which has been used to describe the pharmacokinetic principle to convey nanomedicines to various tumors [1]. As tumors grow, they stimulate the production of new blood vessels or engulf existing blood vessels due to rapid proliferation, which is called tumor angiogenesis. For that reason, tumor vasculature usually has incomplete endothelial lining, causing relatively large pores (0.1–3 μm in diameter) that are different from normal blood vessels, resulting in significantly higher vascular permeability and hydraulic conductivity [2,3,4]. Liposomes, nanoparticles, and macromolecular drugs smaller in size than the pores, therefore, tend to diffuse into tumor tissue via broad pores of tumor vasculature. The size of nanodrugs using the EPR effect plays a crucial role in their efficient transportation [5,6]. It is recommended that the size of these drugs should be over 6–8 nm or more than 40 kDa [7,8,9]. For instance, although nanodrugs vary in size from small molecules to 1–2 μm, extra-small nanodrugs have certain disadvantages, such as being easily extruded by IFP, while extra-large nanodrugs have limitations in terms of low tissue penetration [10,11,12]. Hence, the appropriate size of nanodrugs is important for effective delivery. According to the study by K. Raza et al., most drugs utilizing the EPR effect show optimal uptake efficiency when their size ranges between 100 and 200 nm. Nanodrugs within this size range exhibit better uptake efficiency compared to those smaller than 50 nm or larger than 300 nm [13]. However, several research studies have demonstrated successful transportation of ultrasmall nanoparticles with a size of 2–6 nm and 10 kDa [14,15]. Therefore, while size is the primary determinant for efficient delivery via the EPR effect, other factors such as shape, hardness, charge, and hydrophobicity also contribute to the overall delivery effectiveness. In addition, transported anticancer agents become trapped and accumulate in the tumor site due to the lack of lymphatic drainage by defection of the lymphatic system as the tumor grows [2,16].

The EPR effect is considered a universal pathophysiological phenomenon and has been well observed and documented in solid tumors of rodents, rabbits, canines, and human patients [1,17,18,19,20,21,22,23,24,25]. There are several pathophysiological characteristics of these solid tumors. First, tumors urgently need large amount of nutrients and oxygen, so the tumors have extra-dense and tortuous vasculature with membrane-deficient and fenestrated endothelial structures in some immature vessels, which leads to massive irregular neovascularization in solid tumors with functional and structural abnormalities in tumor blood vessels. The smooth muscle alpha actin of pericytes and smooth muscle cells surrounding the tumor vessels is either dysfunctional or deficient in response to stimuli that regulate blood pressure [26,27,28]. The dominant mechanism of nanoparticle extravasation in tumors is through transendothelial pathways, as recently observed, due to the low frequency of gaps between endothelial cells in tumor vessels [29]. Consistent with this, it has been shown that mature veins or venules, which are constructed by a continuous endothelium with closed interendothelial cell junctions, have a high permeability for macromolecules [30]. These structures make the tumor blood vessels highly permeable to nutrients, particularly macromolecules, enabling their extravasation into the interstitial space of the tumor tissue. Second, the EPR effect in solid tumor tissues is sustained by the coordinated expression of various inflammatory factors, including prostaglandins, bradykinin, nitric oxide, peroxynitrite, interleukin 1 beta, interleukin 2, interleukin 6, proteases, interferon gamma, VEGF, and HIF-1 alpha [23,31,32,33]. Third, the inefficient drainage of the lymphatic system in solid tumor tissue results in the accumulation of extravasated macromolecules in tumor tissues. This provides an opportunity for the passive targeted delivery of macromolecular anticancer drugs [1,23,34].

Due to tumor selectivity of anticancer agents through the EPR effect, several nanomedicines using the mechanism of the EPR effect have been reported and have been in clinical trials or commercialized [35,36,37] (Table 1). For example, Doxil is the first FDA-approved nano-drug, developed using liposomal encapsulation of doxorubicin hydrochloride, resulting in prolonged drug circulation time and high and stable remote loading of doxorubicin [38]. Due to the EPR effect, it is passively targeted to tumors, and its doxorubicin is released to tumor cells through unknown means. In addition, Abraxane, also a targeted nanomedicine through the EPR effect, is a Cremophor-free, 130 nanometer particle form of paclitaxel developed by American BioScience, Inc. It delivers paclitaxel as a suspension of albumin particles in saline, eliminating the need for Cremophor and the associated toxicities, and allowing a shorter infusion duration and use of standard infusion sets [39]. However, although these nanomedicines using the EPR effect for improved delivery efficiency have shown enhanced performance, there are still limitations such as heterogenous efficiency among the patients.

## 2. Differences in Experimental Models and Clinical Models

Since the EPR effect was first discovered over three decades ago, it has been the cornerstone of cancer nanomedicine development. Progress in the field of tumor-targeted drug delivery has been significant, as demonstrated by the approval of several nanomedicinal anticancer drugs. However, the existence and magnitude of the EPR effect, particularly in patients, has recently become a topic of intense discussion. Failures in achieving satisfactory therapeutic benefits in clinical practice have led to debates about the presence and strength of the EPR effect in human solid tumors [3,4]. For example, the EPR effect seems to be more significant in experimental small animal tumor models than in human tumors. When comparing the delivery efficiency of nanoparticles in animal tumor models and human tumors, the latter showed lower values. Nanoparticles are capable of extravasating into tumors not only through gaps between endothelial cells in tumor vasculature but also through transcellular pathways that utilize vesiculo-vacuolar organelles [30,46,47].

The current models used to study the EPR effect in laboratories may not accurately reflect the situation in humans and require reconsideration. While murine tumor models have been widely used, they differ significantly from human tumors in various aspects, such as the rate of development, size relative to host, metabolic rates, and host lifespan. One key difference is the growth rate of tumors, with subcutaneous tumors growing to ±1 cm (≈0.5 g) within 2–4 weeks in mice, equivalent to a ±20 cm (≈1–2 kg) growth in humans [48]. The pharmacokinetics of drug carriers may be significantly affected by the large tumor-to-body weight ratio in mice. In mice, the size of tumors could be increased to as much as 10% of the body weight, whereas an equivalent ratio of the tumor size in a 70 kg human would result in a tumour the size of a basketball [49]. These large tumors filter a significant proportion of the injected drug dose and act as a reservoir, enhancing efficacy while mitigating toxicity. Furthermore, human tumors have significant differences in their microenvironment compared to murine tumors, such as the heterogeneity or absence of fenestrations in the tumor endothelium, acidic or hypoxic areas caused by heterogeneous blood flow through tissues [50], heterogeneous basement membrane (BM) and lower pericyte coverage, and elevation of IFP due to higher density of the ECM. This causes nanomedicines to exit the blood compartment via diffusion instead of the more efficient convective transport (Figure 1) [50,51,52].

Assessing the EPR effect in human tumors is expensive and time-consuming, but understanding the biological factors that contribute to EPR variability is crucial. Tumor histology alone is not enough to predict the EPR effect, so imaging techniques such as CT, MRI, and functional ultrasound are needed to investigate the relationship between vascularization and intratumor accumulation. Alternatively, researchers can use in vivo tumor models, such as patient-derived tumor explant models, which more accurately represent the complexity and heterogeneity of clinical tumors. These models are more mature and less permeable than xenografts [53]. However, the EPR effect of nanoparticles in tumor tissues is highly simplified and overestimated in conventional tumor xenograft animal models established by subcutaneous injection of tumor cells, which do not reflect the heterogeneity and complexity of human tumor tissues [54,55]. Due to the poor similarity of the existing animal models to human cancer, many nanoparticles fail to translate into clinical applications after preclinical testing.

## 3. Challenges in Clinical Translation of the EPR Effect

Although the EPR effect has been significant phenomenon for passive targeted drug delivery, it still has several limitations. The current research on the EPR effect in tumor tissue is highly simplified and overestimated because the tumor xenograft model is mainly established through the hypodermic injection of tumor cells into the mouse flank [54,55]. Unlike the xenograft model, the EPR effect in existing human cancers is highly affected by tumor heterogeneity and the complexity of the tumor microenvironment (TME) [56,57]. The targeting delivery ability of nanoparticles is greatly influenced by factors related to the TME, such as the dense ECM, tumor cell density, hypoxia, and IFP (Figure 2) [58,59]. For example, the dense ECM in patients’ tumor tissue plays an important role as a physical barrier, which prevents upscaled nanostructures from being delivered via intravenous injection [60]. For that reason, passive targeted drug delivery using the EPR effect has critical challenges before overcoming tumor heterogeneity and variation of TME.

Among these points of significance and challenges in concept and application of the EPR effect in human cancer therapy, dense ECM is one of the major obstacles for an efficient EPR effect. The ECM is a significant constituent of tumors, which performs various vital functions such as providing mechanical support, modifying the microenvironment, and serving as a source of signaling molecules [61]. Like other TME conditions, the ECM in tumor tissue has different properties from normal tissue, and these differences have also been investigated for their potential use in tumor-targeted therapy according to previous studies [62,63]. However, unlike the tumor xenograft model, the dense ECM structure in human cancers plays a role as a physical barrier, which blocks the delivery of nanoparticles or antibodies via the EPR effect [64]. To elaborate, in the case of the tumor xenograft model, the drug delivered via intravenous injection penetrates the ECM due to its low density. In contrast, even though targeted anticancer agents arrive around the tumor tissue via leaky blood vessels, they are not able to penetrate the dense ECM to reach the cancer cells. Specifically, the density of the ECM in pancreatic cancer and carcinoma is higher than in other tumor types, making targeted drugs delivered via the EPR effect less effective due to the low ECM penetration rate [65,66]. Therefore, increasing penetration efficiency from blood vessel to tumor tissue is important to overcome the clinical hurdle of the EPR effect.

Another concern for the EPR effect is the roles of IFP [67,68] and solid stress [69] in solid tumor tissue. IFP and solid stress exist due to the dense ECM and the expansion of the tumor mass against surrounding normal tissue [70]. More specifically, because of the dense ECM and the elevated proliferation rate of cancer cells, the IFP of the tumor tissue is higher than that of normal tissue [71]. The IFP of tumor tissue is typically higher than that of normal tissue, ranging from about 5 to 40 mmHg compared to 0–3 mmHg, respectively. This higher IFP at the tumor site is similar to microvascular pressure levels, which can promote the migration of cancer cells and accelerate tumor invasion and proliferation [67,72,73,74,75,76]. Moreover, the high IFP area tends to expand, leading to further complications. As a result, the high level of IFP interferes with the delivery efficiency of targeted drugs via the EPR effect at the tumor site; therefore, it is believed that IFP and solid stress are the major obstacles preventing efficient delivery of macromolecules into tumor tissue [77]. For the transported drug via leaky blood vasculature to be effective, it needs to penetrate and diffuse to the center of the tumor site. However, even if the anticancer agents are well delivered around the tumor site, the drugs are not able to diffuse into the center of the tumor tissue due to the increased IFP of the tumor, which is equal to the surrounding capillary pressure [78,79,80]. For that reason, reducing the IFP of the tumor is key for enhancing the EPR effect. However, another hurdle is that the supply of nutrients is also promoted with tumor growth acceleration [81].

In addition, IFP or solid stress hinders drug penetration into the center of tumor tissue, but it does not prevent the macromolecular agents from extravasating and accumulating in the peripheral area of tumor tissue [82]. As a result, abnormal tumor vasculature, also constructed by tumor angiogenesis and the hypoxic conditions of the tumor tissue, is also an obstacle to the EPR effect [83]. The mechanism of tumor angiogenesis is not clearly understood, which is the reason for unexpected blood vessel growth. However, due to the rapid growth rate of tumor blood vessels, the density of blood capillaries increases, exhibiting a multi-branched morphology and disordered direction [84]. These irregular properties could cause heterogeneity in tumor blood flow, leading to poor blood flow rates at the tumor site. Additionally, various tumor species have a hypoxic condition, which plays a key role in crude blood flow and is one of the TME factors [85]. The hypoxic condition is introduced by specific properties of the tumor, such as tumor angiogenesis and rapid proliferation. Due to cancer proliferation, overexpressed cancer cells consume excess oxygen from tumor vessels [86]. Also, the generation of disordered blood vessels at the edge of the tumor inhibits the delivery of oxygen to the center of the tumor, causing a hypoxic condition in the deeper part of the tumor tissue [85,87,88]. As a result of ineffective blood flow, targeted drugs intended to be delivered via the EPR effect cannot be transported to the tumor tissue, resulting in poor effectiveness [89,90]. It is critical to understand and remember that the peripheral highly vascularized area is the most vigorously growing zone of tumors. The center of tumor tissues lacks blood flow and is necrotic or semi-necrotic.

Nonspecific clearance of nanodrugs by organs such as the liver and spleen has also been a challenge in improving drug delivery efficiency. According to Y. Zhang et al., 30–99% of injected nanoparticles accumulate in the liver after injection, leading to a reduction in drug delivery efficiency [91]. The hepatobiliary clearance of nanoparticles and their interaction with various liver cells, including Kupffer cells, sinusoidal endothelial cells, hepatic stellate cells, hepatocytes, and hepatocellular carcinoma cells, play a significant role in this limitation. To understand the mechanism behind this phenomenon, K. M. Tsoi et al. reported factors such as blood flow dynamics, organ microarchitecture, and cellular phenotype that influence the blood clearance mechanism [92]. However, it is still difficult to overcome this hurdle, so finding a way to avoid non-specific organ accumulation of nanoparticles remains necessary.

## 4. Overcoming the Hurdles of the EPR Effect in Clinical Translation

As reported in the studies above, dense ECM, high IFP levels, and other clinical properties of tumor models derived from them can prevent the delivery of targeted anticancer agents via the EPR effect. To overcome these challenges, various physical and pharmacological strategies have been developed (Figure 3). Ultrasound is one of methods that improves the EPR effect and is highly focused on cancer treatment due to its deep penetration and easy accessibility [93]. Y. Choi et al. introduced high-intensity focused ultrasound (HIFU) to destroy dense ECM and combined it with doxorubicin (DOX) loaded chitosan nanoparticles (CNP) for drug delivery therapy via passive targeting methods [94]. The enhanced therapeutic effect of DOX-CNP was observed in A549 tumor-bearing mice with a rich ECM that were treated with HIFU, due to the successful destruction of the ECM on the A549 tumor site. Similar to the previous study, K. Oh et al. also utilized HIFU to remodel dense ECM and deliver docetaxel-loaded Pluronic F-68 nanoparticles [95]. In addition, S. Wang et al. successfully transported a cancer-specific antibody to a tumor that had been pre-treated with HIFU [96]. They used A431 human epidermoid carcinoma cells which expressed the Lewis^y^ antigen, commonly found in most human carcinomas. They also labeled the anti-Lewis^y^ B3 with Alexa-647 or ^90^Y to analyze the penetration and accumulation level of the antibody at the tumor site. They demonstrated that HIFU treatment induced an increasing gap size between fiber bundles and disruption of the ECM, which enhanced the distribution of labeled antibody and therapeutic effect on A431 tumor-bearing mice.

Reducing IFP is also a key strategy for overcoming tumor heterogeneity, which interferes with the EPR effect. C. G. Willett et al. treated patients with primary and nonmetastatic rectal cancer with the VEGF-specific antibody bevacizumab alone to evaluate its effects [97]. VEGF plays an important role in tumor angiogenesis, and due to that property, VEGF blockade has been tested in clinical trials [98,99]. They observed a significant reduction in IFP levels from 15 to 4 mmHg when treated with the VEGF antibody. This reduction in IFP was inferred to be due to the normalization of tumor blood vessels based on several analyses [100]. Moreover, Y. Fan et al. used imatinib (IMA) to reduce IFP and enhance the delivery effect of liposomes to solid tumors [101]. IMA, which targets the BCR-ABL gene, also has the potential to inhibit the tyrosine-protein kinase Kit and PDGF receptors (PDGFR) [102]. The PDGF ligand-receptor system plays a significant role in tumorigenesis, and therefore the IFP level could be reduced by inhibiting PDGFR [103]. For the purpose of reducing IFP using IMA, Y. Fan and the team loaded IMA into stabilized liposomes. They reported a great reduction of IFP in B16 melanoma, with improved drug delivery effects and demonstrated enhanced accumulation of doxorubicin-loaded liposome by an enhanced EPR effect. Also, to reduce IFP level, collagenase was introduced by M. Kato and colleagues [104]. Type I collagen is a major component of ECM in tumors, which is one of the causes of high IFP by maintaining dense ECM. Type I collagenase, also called matrix metalloproteinase-1, can degrade type I and III collagen. For that reason, M. Kato et al. injected type I collagenase intravenously and showed enhanced accumulation of liposome/pDNA complex.

Hyperthermia therapy (HT) has also been used for cancer therapy to improve the EPR effect as a physical treatment. Although HT is usually used in combination with other cancer therapies, such as chemotherapy or radiotherapy, it can be applied broadly due to its wide range of applications [105]. The simple mechanism of HT to improve the EPR effect is by increasing blood flow to the tumor tissue, resulting in improved drug delivery efficiency and oxygenation to solve the hypoxic condition [106,107]. For example, G. Kong et al. reported that HT could promote liposome accumulation at tumor sites [108]. They investigated the efficient range of temperatures for the best quality of therapy and demonstrated that the accumulation rate of liposomes around 100 nm was enhanced in human ovarian cancer (SKOV-3) at 42 °C. They also set up the HT schedule with an 8 h interval for several reasons, including thermotolerance. Moreover, in recent times, HT has been used for immunotherapy [109]. HT is known to cause immunogenic cell death, resulting in the generation of damage-associated molecular patterns including high mobility group B1, extracellular adenosine triphosphate and surface-expressed calreticulin. For using this, Y. Guo et al. have developed a photothermally triggered immunotherapeutic strategy using magnetic-responsive immunostimulatory nanoagents (MINPs) loaded with superparamagnetic iron oxide nanoparticles and cytosine-phosphate-guanine oligodeoxynucleotides for imaging-guided photothermal destruction of primary tumors, releasing tumor-associated antigens and activating antitumor immune responses [110]. The MINPs acted as a contrast agent for bimodal imaging and a magnetic-targeting therapeutic agent under external magnetic fields. The study demonstrates a potential individualized diagnosis and therapy for various tumors with high specificity, maneuverability, and biocompatibility.

**Figure 3 ijms-24-10082-f003:**
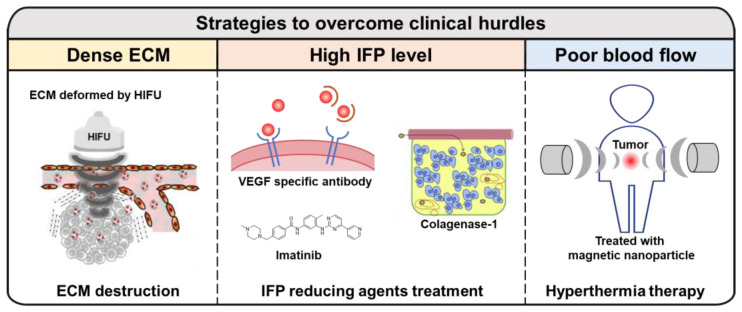
Strategies to overcome clinical hurdles of the EPR effect. There are challenges of the EPR effect in clinical models, such as (i) dense ECM; (ii) high level of IFP; (iii) poor blood flow. To overcome these limitations, several studies have been reported, using HIFU, IFP reducing agents, and HT. Reproduced by permission [94,104].

Understanding the differences between clinical translation models and experimental models is one of the keys to overcoming challenges in the EPR effect. The hurdles of the EPR effect, which are described in previous chapter, commonly arise from the specific characteristics of clinical models. For that reason, the actual tumor model should be figured out. However, due to the difficulty of direct experimentation on patients, research into the EPR effect in clinical models has not been going well. To solve this problem, S. Jeon et al. evaluated the tumor-targeting efficiency of glycol chitosan nanoparticles (CNPs) in different pancreatic tumor models, including a pancreatic cancer cell line, patient-derived cancer cell, and patient-derived xenograft models (Figure 4) [111]. They found that differences in the tumor tissues greatly affected the tumor-targeting efficiency of CNPs and that patient-derived xenograft models could mimic inter-patient tumor microenvironments, which could predict the response of various nanoparticles in individual tumors for personalized cancer therapy. Based on this study, patient-derived xenograft models could be used for personalized tumor therapy to overcome tumor heterogeneity observed among different patients. In other ways, M. Haque and colleagues mimicked patient-derived tumor models and TME using patient-derived organoids (PDOs) and stromal cells [112]. They developed a tissue-chip model that improves long-term cell survival of PDOs and incorporates stromal cells to recapitulate the tumor microenvironment of PDAC. The model allows for drug testing and evaluation of anticancer drugs before application in the clinic. This study confirms the importance of stromal cells in PDAC aggravation and the potential for cancerous epithelial cells to manipulate stromal cells to create a favorable TME. The platform can be expanded to include additional components to better recapitulate the complex PDAC TME and advance personalized medicine in PDAC.

Other strategies for improving the EPR effect have been introduced by several researchers. For example, R. J. Christie and colleagues suggested the use of poly(ethylene glycol)-block-poly(L-lysine) modified with 2-iminothiolane and cyclo-Arg-Gly-Asp peptide with encapsulation of siRNA for transportation to solid tumors [113]. The cyclo-Arg-Gly-Asp peptide tends to bind with overexpressed αvβ3 and αvβ5 integrin receptors, so cyclo-Arg-Gly-Asp modified poly(ethylene glycol)-block-poly(L-lysine) showed better accumulation at the tumor site than unmodified poly(ethylene glycol)-block-poly(L-lysine). To avoid nonspecific clearance by liver accumulation, A. Dirisala et al. developed a method to selectively coat liver sinusoids using poly(ethylene glycol) (PEG)-conjugated oligo(l-lysine) (OligoLys) to enhance the delivery of nanomedicines [114]. They found that PEG-OligoLys specifically attached to liver sinusoids, leaving other tissues uncoated. Two-armed PEG-OligoLys was cleared from sinusoidal walls to the bile, while linear PEG-OligoLys persisted in the sinusoidal walls. Also, B. Ouyang et al. discovered that a dose threshold of 1 trillion nanoparticles in mice significantly improved the delivery of nanoparticles to tumors [115]. Beyond this threshold, the uptake of nanoparticles by Kupffer cells was overwhelmed, leading to reduced liver clearance and prolonged circulation, resulting in increased delivery of nanoparticles to tumors. This approach achieved up to 12% tumor delivery efficiency, targeting 93% of tumor cells and enhancing the therapeutic effectiveness of Caelyx/Doxil.

## 5. Conclusions

Blood vessels at a tumor site have relatively wide pores compared to those in normal tissue. This phenomenon, known as the EPR effect, allows targeted drugs ranging from 10 nm to 300 nm to easily accumulate at the tumor tissue via diffusion from tumor blood vessels into tissue [116,117,118]. The EPR effect has been a promising method for drug transportation to the tumor tissue, and various nanoparticles have been developed to transport anticancer agents via this effect with different physical and chemical properties, enhancing efficiency.

However, despite the development of nanoparticles, many clinical trials have failed due to the specific properties of clinical models [119,120], which differ from tumor xenograft models. The main challenges have arisen from tumor heterogeneity in the TME across various species and types of tumor tissues, as well as a lack of understanding of these heterogeneous properties. Nevertheless, to overcome these challenges, several studies have reported improvement of drug delivery models via the EPR effect, such as degradation of the ECM and reduction of IFP levels. By producing and analyzing actual clinical tumor models in collaboration with various fields, it is expected that the EPR effect will be improved by controlling the environment while gaining essential understanding of the tumor microenvironment. Moreover, we expect that enhancing drug delivery in clinical models via an improved EPR effect will be key for targeted cancer therapy.

## Figures and Tables

**Figure 1 ijms-24-10082-f001:**
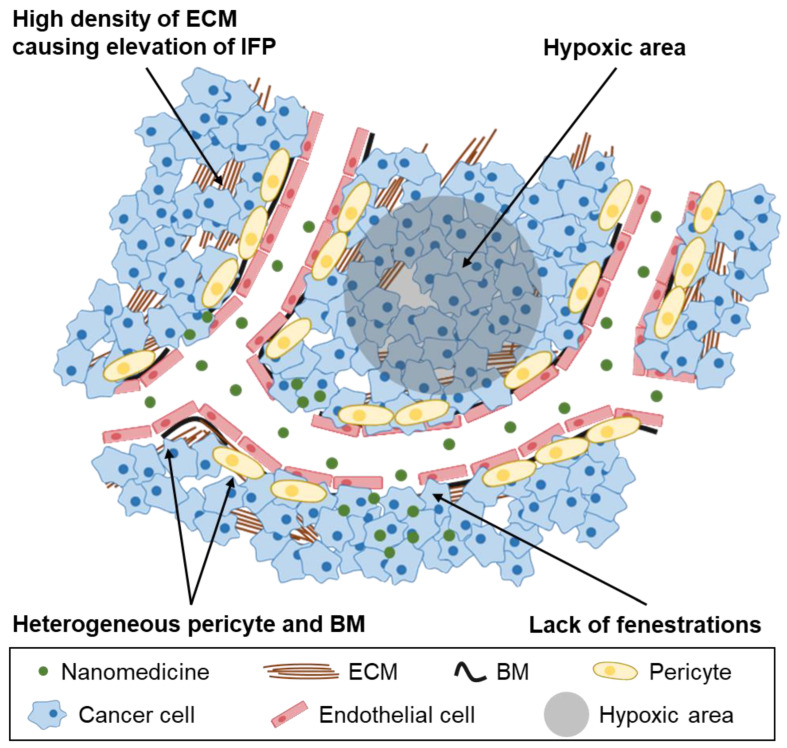
Schematic depiction of nanomedicine transportation via the EPR effect in human tumors. Human tumors have significant differences in their microenvironment compared to murine tumors, such as the heterogeneity or absence of fenestrations in the tumor endothelium, acidic or hypoxic areas caused by heterogeneous blood flow through tissues, heterogeneous BM and lower pericyte coverage, and elevation of IFP due to higher density of the ECM.

**Figure 2 ijms-24-10082-f002:**
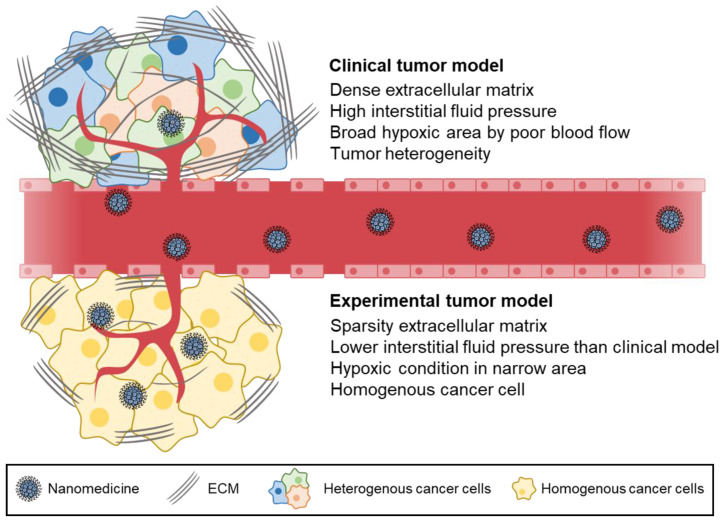
Schematic illustration shows the differences between clinical and experimental tumor models. In the case of a clinical translation model, there are several properties that are different from a xenograft model, such as a dense ECM and high IFP, which can present challenges to efficient drug delivery by the EPR effect.

**Figure 4 ijms-24-10082-f004:**
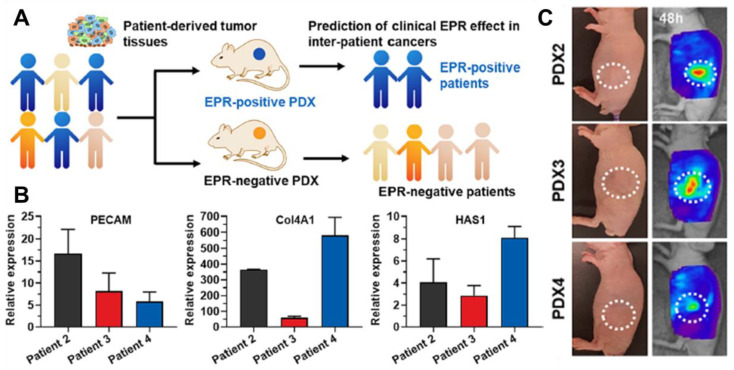
(**A**) Schematic illustration of the identification of EPR-positive patients via experiments using patient-derived xenograft models. (**B**) Different expression levels of PECAM, Col4A1, and HAS1 were shown among the patients. (**C**) Different chitosan nanoparticle uptake levels were shown among the patients. According to the study, the EPR-positive patients could be identified based on the results. The white circle indicated the position of the explanted tumor. Reproduced by permission [111].

**Table 1 ijms-24-10082-t001:** List of nanomedicines using the EPR effect as a therapeutic mechanism.

Name	Anticancer Agents	Target Disease	FDA Approval Year	Diameter (nm)
Doxil [38]	Doxorubicin	Ovarian cancer Kaposi’s sarcoma	1995	<100
Abraxane [39]	Paclitaxel	Breast cancer Non-small cell lung cancer Pancreatic cancer	2005	<130
Myocet [40]	Doxorubicin	Breast cancer	2000	<150
Marqibo [41]	Vincristine	Philadelphia chromosome-negative lymphoblastic leukemia	2012	<100
Onivyde [42]	Irinotecan	Metastatic pancreatic cancer	2015	<130 [43]
Vyxeos [44]	Daunorubicin Cytarabine	Acute myeloid leukemia	2017	<100 [45]

## Data Availability

Not applicable.

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
