# Peer review of "Perspectives for Improving the Tumor Targeting of Nanomedicine via the EPR Effect in Clinical Tumors"

_ijms, 2023, doi:10.3390/ijms241210082_

Round 1

Reviewer 1 Report

The manuscript “Perspectives for improving the tumor targeting of nanomedicine in clinical tumors” after presenting the basics of EPR mechanism in nanomedicine with pros and cons, discusses some strategies recently proposed to overcome the critical points of this effect that limit or prevent the clinical translation of nanoformulations with EPR based tumor targeting.

The topic of this review article is interesting and it is clearly written, but, in my opinion, one key point should be definitly addressed before publication:

1. the paper deals ONLY with EPR mechanism while the present title describes a much more general topic, the title should be changed stating clearly the focus of the review and indicating ‘improving the EPR tumor targeting’

2. a fundamental point is completely missing throughout the whole review: the size of the described nanoformulation and the effect of the size of the nanomedicine on the EPR effect. In my opinion it is a point that can not at all be completely neglected in a review on the EPR mechanism. Even if it is not the focus it is essential to mention it and discussed the influence that the size could have related to the different limitation of the EPR . I think that a dedicated paragraph with suitable references should be added or, at least, the topic should be discussed when pertinent in different points of the review. There are recent reviews that should also be cited, in my opinion,  (two of them could be Polymers  202214(13), 2601;  https://doi.org/10.3390/polym14132601; Theranostics 2020; 10(3):1319-1331. doi:10.7150/thno.37543. https://www.thno.org/v10p1319.htm ) that consider also this point.

3. in table 1, where the clinically approved nanomedicine are listed, a column with the type of particle used and their size should be added in order to help the reader to understand better and also for completeness of information

Reviewer 2 Report

M.K. JooK. Kim, and their team presented a manuscript entitled “Perspectives for improving the tumor targeting of nanomedicine in clinical tumors”. The authors mainly focused on the basic mechanisms and challenges associated with the enhanced permeability and retention (EPR) effect of nanocarriers for the effective delivery of anticancer drugs. In addition, the team also focused on EPR beyond targeting strategies to overcome the hurdles associated with the clinical translation of nanomedicines.

Overall, the manuscript is fascinating and can be considered for a possible publication in MDPI IJMS. However, we recommend the authors address the following issues to reach broad audiences of different disciplines. 

1.    Abstract, Line 17:  high full name (IFP)? à high interstitial fluid pressure (IFP)

2.    The manuscript needs English editing.

For example, Line 49: tumor vascular? à tumor vasculature.

Line 194: leaky blood vascular to be effective. à leaky blood vasculature to be effective.

3.    Lines 103-105: Please cite the original article that suggested the vesiculo-vacuolar organelles (VVOs)as the transcellular pathway for effective extravasation of nanoparticles (Am J Pathol 1988; 133(1): 95-109 and Microsc Res Tech 2002;57(5):289-326).

4.    Please give the abbreviations and acronyms after the first description in the manuscript.

5.    Please remove the abbreviations and acronyms if the description is given only once. These abbreviations and acronyms will not provide any scientific contribution or discussion. Instead, they will distract the readers and increase the count of words. 

For example, 

Line 105: (VVOs)

Line 279: (ICD)

Line 280: (DAMPs)

Line 281: (HGMB1)

Line 281: (ATP)

Line 282: (CRT)

Line 284: (SPIO)

Line 284: (CpG ODNs)

Line 251: (c-Kit)

Line 260: (MMP-1)

6.    Abbreviations or acronyms were described many times.

For example,

Lines 151, 159, and 177: extracellular matrix (ECM)

Lines 12, 93, and 178: enhanced permeability and retention (EPR)

Lines 120 and 141: basement membrane (BM) 

7.    Don’t use abbreviations or acronyms if used a few times. Use the patient-derived tumor explant model instead of its abbreviation in Figure 4.

8.    We recommend that the authors describe active targeting strategies as EPR beyond approach. For example, the targeting moieties are decorated to the surface of the nanocarriers, preferably at the distal end of the polymers that form the shell of nanocarriers. For example, polyion complex micelles prepared from cyclic RGD (Arginine-Glycine-Aspartic acid) peptide ligand-installed poly(ethylene glycol)-poly(L-lysine-thiol) block polymers and plasmid DNA encoding soluble form of human vascular endothelial growth factor receptor-1 (VEGFR-1) (or soluble fms-like tyrosine kinase-1: sFlt-1) selectively targeted vβ5 and vβ3 integrin receptors overly expressed on cancer cells and tumor vascular endothelial cells upon intravenous injection. The targeted tumor cells expressed sFlt-1 protein, a potent antiangiogenic protein, which efficiently captureVEGF, inhibited new vasculature formation (anti-angiogenesis), and ultimately showed antitumor activity. Similarly, integrin-targeted cRGD-decorated polyplex micelle loading siRNA targeting VEGF-receptors successfully knocked out the expression of VEGF receptors, thereby preventing the neovasculature formation in tumors (ACS Nano 2012;6(6):5174-89). 

9.   We recommend that the authors describe the most significant hurdle for the clinical translation of tumor-targeting nanocarriers and recent RES blockading strategies for effectively targeting nanocarriers to the tumor site. One of the essential critical hurdles to the clinical translation of systemically administered nanocarriers (pseudo-stealth nanocarriers) is nonspecific uptake by the reticuloendothelial system (RES) organs, most notably the liver, because it not only substantially decreases the dose of nanocarriers to the diseased tissue but also can raise toxicity concerns (Nat Mater 2016;15(11):1212-1221). Transient blockading of the scavenging functions of RES received great attention to prolong the blood circulation longevity of nanocarriers, thereby increasing the chances of accumulating nanocarriers at the targeted tumor site (Sci Adv 2020;6(26):eabb8133Nat Mater 2020;19(12):1362-1371, and Nat Biomed Eng 2020;4(7):717-731). 

English language editing is recommended. 

Round 2

Reviewer 1 Report

In the present version the authors have improved the points that I had signalled.

I have only two very minor issues:

1. In the table I suggest do put diameter (or radius?) on the top of the column and the units in within brackets like ‘diameter (nm)’ to better specify if it is a diameter or a radius;

2. I found the English of the added parts less fluent than the other parts of the text, the construction of the added sentences should be reconsidered in some point. For example, the sentence ‘However, it has been still difficult to overcome this hurdle, so the way of avoiding liver accumulation of nanoparticle is still on demand’ should be rewritten, in my opinion, the verb tense does not sound correct to me and also the meaning of ‘on demand’ is out of context here. There are also other sentences with similar problems.

2. I found the English of the added parts less fluent than the other parts of the text, the construction of the added sentences should be reconsidered in some point. For example, the sentence ‘However, it has been still difficult to overcome this hurdle, so the way of avoiding liver accumulation of nanoparticle is still on demand’ should be rewritten, in my opinion, the verb tense does not sound correct to me and also the meaning of ‘on demand’ is out of context here. There are also other sentences with similar problems.

Reviewer 2 Report

Accepted 
